# The Effectiveness of Pharmacological Synchronization of the Estrous Cycle in Hinds (*Cervus elaphus* L.): A Pilot Field Trial

**DOI:** 10.3390/ani10112148

**Published:** 2020-11-19

**Authors:** Anna J. Korzekwa, Władysław Kordan, Angelika M. Kotlarczyk, Roland Kozdrowski

**Affiliations:** 1Department of Biodiversity Protection, Institute of Animal Reproduction and Food Research of Polish Academy of Sciences, Tuwima 10 Str., 10-748 Olsztyn, Poland; a.kotlarczyk@pan.olsztyn.pl; 2Department of Animal Biochemistry and Biotechnology, University of Warmia and Mazury, Oczapowskiego 5, 10-718 Olsztyn-Kortowo, Poland; wladyslaw.kordan@uwm.edu.pl; 3Faculty of Biological and Veterinary Sciences, Nicolaus Copernicus University, Gagarina 11, 87-100 Toruń, Poland; roland.kozdrowski@umk.pl

**Keywords:** red deer, estrous cycle, synchronization, ovarian steroids, insemination

## Abstract

**Simple Summary:**

Reproductive biotechnics of Cevidae species may increase valuable traits for the agricultural sector, and improve genetic diversity in relatively small farmed populations. The first step of biotechnics in terms of suitability for embryo transfer or insemination is preparation of females for proper phase of the estrous cycle. Based on available synchronization protocols applicated in red deer and domestic ruminant females we estimated the protocol for hinds in view of future routine use. Progestin analogues in the form of an insert (CIDR) or a sponge (Chronogest) and injection of equine chorionic gonadotropin (eCG; Folligon) were used on the appropriate day for inducing the estrus. The effect of synchronization was checked by measurement of progesterone (P4) and 17-beta estradiol (E2) concentration by RIA, pregnancy-associated glycoproteins (PAG)s concentration by EIA and insemination effectiveness—pregnancy rate. The better method of the estrous cycle synchronization found down to be progestin sponge and eCG based protocol in hinds considering hormone responsiveness and pregnancy rate.

**Abstract:**

The aim was to estimate the effective pharmacological method of the estrous cycle synchronization by checking the effects of synchronization by measurement of progesterone (P4) and 17-beta estradiol (E2) concentration by RIA and artificial insemination. The experiment was performed at the red deer farm in Rudzie (North-East Poland; 3 year’s old). The herd (N = 14) was kept away from bulls and was divided in two groups of seven animals. In the Group I, CIDR insert (0.3 g of P4) was applicated intravaginally for 12 days; a second insert replaced the first one for the next 12 days, and next 200 IU of equine chorionic gonadotropin (eCG) was injected intramuscularly (Folligon). Estrus was expected 48 h after eCG injection. In the Group II, Chronogest sponge (20 mg of flugestone acetate) was applicated intravaginally and after 7 days replaced with second chronogest sponge for 7 days. After removing the sponge, on the same day eCG was injected and estrus was expected after 48 h. Artificial insemination was provided with frozen-thawed semen twice: 12 and 24 h after expected estrus. The peripheral blood from the jugular vein was collected each time when the inserts or sponge were applicated and 40 days after insemination. The concentration of P4 and E2 in plasma was measured by RIA. The effectiveness of insemination was monitored by *pregnancy-associated* glycoproteins determination and observed by the number of calves born. Two pregnancies were confirmed in Group I and five in Group II based on PAG concentration. One newborn was observed in Group I and five in Group II. Both methods of synchronization are effective in hinds based on the received profile of steroids. Although the sponge shape in case of chronogest is better comparing with CIDR, which was not completely deposited in the vagina of hind, potentially leads to bacteria inflammation, and it disturbs the rightful endocrine regulation. Moreover, pregnancy rate and hormone responsiveness were better in Group II.

## 1. Introduction

Red deer (*Cervus elaphus* L.) belongs to the most abundant cervid species in Europe. Wild living and farming populations of red deer have an increasing economic, cultural, and ecological importance [1]. Red deer are a seasonally reproduced ruminant, and females exhibit several estrous cycles in autumn [2,3]. Seasonal breeding in ruminants ensures the birth of offspring under the most favorable food and climatic availability conditions [4]. We have recently published the results concerning red deer bull semen cryopreservation and the suitability of oocytes for fertilization in vitro [5]. With the valuable cryopreserved semen at our disposal, in this study we intend to develop a method of pharmacological hind synchronization that is effective in the context of insemination and pregnancy development.

Reproductive biotechnics including artificial insemination are common routine in domestic ruminant management but have also been developed in Europe during the last 20 years in red deer [6,7,8]. Deer breeding is focused on animal selection towards obtaining the high meat quality. Males are preferred for trophy hunting and their ability to pass valuable functional features in the next generations. The increased pregnancy rates of the hinds entering puberty and an improvement in their lactation performance are associated with the improvement of the farmed deer reproductive performance [9]. Thus, the offspring with favorable features and high efficiency of fertilization oriented toward the acceleration of their genetic improvement rates are a result of earlier selection of high-indexing parents.

Considering the synchronization protocol in cows, representing domestic ruminants whose reproductive seasonality is not dependent on the light season, most often protocols are based on application of progesterone (P4) and prostaglandin (PG)F_2α_ analogues [10]. Whereas in domestic ruminants seasonally reproduced, such as ewes or goats, melatonin in photoperiod-dependent breeding animals is considered as factor which controls reproduction, inducing changes in the annual pattern of reproduction and the perception of photoperiod. Induction of estrus by application of hormones has allowed increased use of artificial insemination in small ruminants, in which there is the difficulty of detecting estrus [11]. Synchronization of estrus possesses commercial potential and allows control of lambing, with subsequent synchronization of weaning of young animals. Sheep and goats are short-day ruminants; their reproductive activity takes place during day fall. Melatonin can be considered as pro-gonadotrophic stimulus in these species. Estrus and ovulation may be induced during the anestrus period by hormonal treatments. Moreover, inducers of LH release have to be associated with a progestagen treatment; eCG is still the most efficient inducer of LH release [11]. Fletcher [12] described that natural estrus is not easily detected on commercial deer farms, where hinds are naturally synchronized.

Although a synchronization program in farm ruminants is commonly used and a synchronization protocol has been well established, in cervids there are only a few papers describing the method of pharmacological estrus synchronization and specific time interval from eCG injection and effective insemination method simultaneously. So far, a P4-impregnated controlled internal drug releasing (CIDR) device was used as a pharmacological synchronization in red deer, inserted intravaginally for 12 days and an intramuscular injection of 200–250 IU of equine chorionic gonadotrophin (eCG) given at the time of CIDR withdrawal. Hinds exhibited estrus between 36 and 60 h later and ovulated 24 h after the onset of estrus [13,14]. However, time of estrus and ovulation in those protocols differ. The establishment of effective synchronization protocol in hinds will be helpful for further reproductive biotechnics realization and primarily it supports successful fertilization in red deer. Based on the available synchronization protocols for ruminants and taking into account the reproductive seasonality of red deer, the aim of the study was to develop an effective protocol for the synchronization of the estrous cycle in hinds, confirmed by the measurement of ovarian steroids, *pregnancy-associated* glycoproteins (PAG)s concentration in blood plasma, and the number of calves born after insemination.

## 2. Materials and Methods

### 2.1. Hinds Selection for Synchronization of the Estrous Cycle

Fourteen healthy 3-year-old red deer females were used for the estrous cycle synchronization and artificial insemination at a deer farm in Rudzie near Gołdap (North-East Poland). Each of the selected hinds was once pregnant and gave birth to a healthy calf. The induction of estrus and ovulation in hinds was performed beginning from 1 September 2019. We received the agreement of Local Ethical Committee in Olsztyn (Poland, Agreement Number 7/2019).

### 2.2. Evaluation of the Effective Protocol of the Estrous Cycle Synchronization in Hinds

Seven hinds estimated as Group I were synchronized by applying a double controlled internal drug-release (CIDR) insert (type G, Zoetis, New York, NY, USA; 0.3 g of P4), using a 12-day regimen of intravaginal CIDR devices inside the cervix. For better synchronization, the device was replaced after 12 days to maintain the luteal concentration of P4 until the end of the treatment period. Additionally, 200 IU of eCG (Folligon, *Intervet,* International B.V., Boxmeer, Holland) was injected intramuscularly on day 1. The estrus was considered to occur 48 h after the second CIDR insert removal in the hinds.

Group II consisted of seven hinds, which were synchronized by double application of Chronogest sponge inside the cervix (20 mg of flugestone acetate; MSD Animal Health) for 7 days and next 200 IU of eCG was injected intramuscularly (Folligon). Estrus was expected 48 h after eCG injection. Detailed protocol of synchronization is presented as Scheme 1.

Each time when application of insert or sponge was provided and at the time of expected estrus, simultaneously the blood from jugular vein was collected for determination of E2 and P4 concentration in the blood plasma using radioimmunoassay (RIA). For exclusion of bacteria inflammation, cervical swabs were collected. PAGs concentration was determined in blood collected during CIDR or Chronogest application and from hinds in expected pregnancy (40 days after insemination) from Group I and Group II. Letters A–E determine time points when blood was collected.

### 2.3. Semen Collection, Cryopreservation, and Preparation for Artificial Insemination

The epididymal semen was collected from seven wild bulls shot during the hunting season post-mortem (September/October 2019) in Strzałowo Forestry (7–8 years old, based on antler shape and clash of teeth, North-East, Poland). The frozen-thawed semen from the same bull was used throughout the experiment. The commercial extender Bioxcell (IMV) was used for dilution of the semen. The concentration of fresh spermatozoa in the ejaculates was 6.93 × 10^9^/mL, and the motility was 95% (CASA; Hamilton-Thorne Biosciences, Beverly, MA, USA). The spermatozoa were cryopreserved according to Korzekwa et al. [5]. Briefly, sperm were loaded into straws and frozen in nitrogen vapor at the concentration 100 × 10^6^/mL, and its post-thaw motility was 75%.

### 2.4. Artificial Insemination

After thawing, the semen was kept in water bath for 2 min at 35–37 °C and deposited into cervix 12 h after expected estrus. The second insemination was made 12 h later. Artificial insemination (AI) gun (COBA-Select Sires, Tyler, TX, USA) and standard sheath containing a straw of frozen–thawed red deer semen and a capillary tube guide was inserted into the cervix through the speculum. An attempt was then made to manipulate the capillary tube and AI gun through the cervix into the uterus. Semen was deposited just as the AI gun passed through the last annular ring and entered the uterus. Forty days post insemination the peripheral blood from jugular vein was collected for determination P4, E2 by RIA and PAGs concentration determination examined by EIA according to Korzekwa et al. [15].

### 2.5. Determinations

#### Steroid and PAG Concentration

The plasma measurements of P4 and E2 were performed using RIA (KIP1458 and KIP0629, DIAsource ImmunoAssays, Louvain-la-Neuve, Belgium). For P4, the standard curve values ranged from 0.12 to 36 ng/mL, and the ED50 of the assay was 0.06 ng/mL. The intraassay and interassay coefficients of variation (CVs) were 6.5% and 8.6%, respectively.

For E2, the standard curve values ranged from 1 to 355 pg/mL, and the ED50 of the assay was 0.5 pg/mL. The intraassay and interassay CVs were 4.7% and 6.1%, respectively.

All samples were pipetted in duplicate for each run, and validation was performed according to Korzekwa et al. [5,15].

The concentration of PAGs in the serum was examined by EIA Pregnancy Test (99-41169, IDEXX, Westbrook, ME, USA) after validation according to Korzekwa et al. [5]. The intra- and interassay CVs were on average 8.0% and 7.9%, respectively. According to the given formula-determined quantity, the test confirmed the presence or absence of PAGs in each sample. A circulating concentration was calculated based on absorbency, where an absorbency of PAGs below 0.3 was considered nonpregnant and above 0.3 as pregnant.

### 2.6. Statistical Analyses

The statistical analysis of the results was performed using GraphPad Prism (GraphPad PRISM, Version 8.3.0, San Diego, CA, USA). The relationship between concentration of P4 and E2 was determined using two-way analysis of variance (ANOVA) followed by a Bonferroni post hoc test. The statistical analysis involved a comparison among Group I and Group II between each time point of synchronization protocol. The difference between concentration of PAGs in nonpregnant hinds from Groups I and II (blood collected during change of CIDR insert or Chronogest sponge) and pregnant animals was determined using one-way ANOVA followed by Bonferroni test. All numerical data were expressed as the arithmetic mean ± SEM. Differences at *p* < 0.05 were considered statistically significant.

## 3. Results

### 3.1. Concentration of Progesterone and 17-Beta Estradiol in Blood Plasma

#### P4 Concentration

Concentration of P4 in the blood plasma in Group I was increased after the first CIDR insert application (B; *p* < 0.05) comparing with anestrus, ovulation, and pregnancy (A, D, and E). The second replacement of CIDR (C) was not effective because the concentration of P4 was at this time similar to other time points (Figure 1a; A, D, and E; *p* > 0.05).

Double Chronogest sponge application in Group II elevated the concentration of P4 comparing with anestrus and ovulation (A, D; *p* < 0.05). Moreover, the increase of P4 concentration to the similar level as during application of sponges was observed in pregnancy (E; *p* < 0.05).

Statistically differences were observed between Group I and Group II at the time of Chronogest sponge or CIDR insert replacement (C; *p* < 0.001) and in pregnancy (E; *p* < 0.05) and the level was higher in Group II (Figure 1a).

#### E2 Concentration

E2 concentration in the blood plasma in Group I was increased after first CIDR insert application (B; *p* < 0.05) comparing with other time points (Figure 1b; A, C–E).

Twice Chronogest sponge application and Folligon injection for inducing estrus in Group II elevated the concentration of E2 comparing with anestrus and pregnancy (A, D; *p* < 0.05).

Statistically differences were observed between Group I and Group II at the time of Chronogest sponge or CIDR insert replacement, respectively (C; *p* < 0.01), and during insemination (C, D; *p* < 0.00001) and the level was higher in Group II (Figure 1b).

### 3.2. Evaluation of the Effectiveness of Pharmacological Synchronization of the Estrous Cycle in Hinds

PAGs presence was detected in the serum of two individuals from Group I and of five individuals from Group II on 40th day of pregnancy. Whereas pregnancy was not confirmed in all hinds, during the steroid analysis, five females were excluded from Group I and two were excluded from Group II for the time period—pregnancy (Figure 1).

One newborn was observed in Group I (fertility index: 14%) and five in Group II fertility index: 71%). There was no inflammation in any swabs. Although, before insert change procedure or directly before its removal from cervix in all individuals from Group I we observed that not all lengths of CIDR insert were deposited in the cervix.

## 4. Discussion

Our results showed that synchronization protocol of hinds based on double Chronogest sponge application with seven days intervals and combination with Folligon injection is the most effective. Moreover, we established the exact time points in this protocol, which affect the high index of pregnancy and the same index of live birth 71.42% (five newborns for seven inseminated females). We also conducted a similar experiment in which the hinds were synchronized through the application of the CIDR insert twice at 12-day intervals, followed by Folligon injection after the CIDR device withdrawal (not published data). The difference was the method of insemination. On the contrary to artificial insemination, those females were inseminated by laparotomy with fresh semen that occurred 62–64 h after eCG injection. Despite the relatively high number of animals in this experiment—40 animals—insemination with fresh semen, which compared to cryopreserved semen had better parameters and the deposition of semen was directly in the uterus, the pregnancy rate, measured by the concentration of PAGs and the observation of calves, was only 40%. Our observation concerning deposition of CIDR insert in cervix of females was that not all of the insert was placed inside the cervix and it may influence disturbances in hormone secretion to the reproductive tract during the synchronization by CIDR in the case of hinds. It was also a risk of inflammation from the side of long term open cervix, the consequence of which was failure in fertilization. It is obvious that the weight of hind correlate with the size of the uterus and cervix. In our case the weight of all experimental females was 85–90 kg. We speculate that CIDR inserts would be more appropriate for lambs or cows. Moreover, the effectiveness of estrus synchronization demands careful determination of the time interval between eCG injection, estrus, and insemination. Little data regarding artificial insemination based on detected estrus are available in cervids. Indeed, hinds show little or no overt physical signs of estrus (e.g., swelling of the vulva), and hind mounting of other hinds is not common (in contrast to cattle) [16]. We also did not observe any visible signs of rut in the experimental hinds. Moreover, in the case of cows, the usefulness of ultrasound examinations in the control of follicular development and ovulation is often practiced and we are considering it in continuation of our studies.

The results of Anel-Lopez et al. [17] showed the correlation between time period from Folligon injection and insemination, which strictly influence the fertility. The highest index of fertilization was obtained after 53–53:30 h post eCG administration using the same concentration of semen (100 × 10^6^/mL). In our case, the time period from Folligon administration and insemination was longer: 60–62 h. Our highest fertility index was 71.42% and it was lower than the one obtained by Anel-Lopez et al. [17], almost 78%. The number of animals in our experiments was smaller and the synchronization should be continued on a larger experimental group but both our and the results of Anel-Lopez et al. indicated that, in the case of red deer females, time period from eCG should be no longer than 62 h or even shorter during synchronization [17]. The levels of P4 and E2 (Figure 1) received throughout the synchronization were the highest in the case of hinds synchronized by Chronogest and Folligon (Group II), which indicated better responsiveness for this synchronization protocol [18]. Moreover, it should be mentioned that only two pregnancies were diagnosed by PAGs concentration in hinds synchronized with CIDR application and one calf was born in this experimental group. The reason may be worse hormone balance in the prefertilization period comparing with the second experimental group, which demands further study. Considering the effectiveness of pharmacological synchronization of the estrous cycle in hinds in terms of suitability for insemination, but also in vitro fertilization, we will continue the experiments by evaluation of luteinizing hormone (LH), anti-Müllerian hormone (AMH), and follicle stimulating hormone (FSH) concentration in plasma [19,20,21]. These parameters may answer the question about the ovarian follicle development and time of ovulation, which is crucial in proceeding of the reproductive biotechnics in red deer also as a model for reintroduction in cervids at risk of extinction [22,23]. Moreover, the receptivity of the uterus for steroids and other factors supporting the pregnancy development will be the subject of our research.

## 5. Conclusions

In summary, we showed that pharmacological synchronization approach by implementation of Chronogest and Folligon is better than CIDR and Folligon in terms of the effectiveness of fertilization rate in hinds. Artificial insemination using cryopreserved semen from valuable bulls and possibility of hind herd synchronization without the presence of bull possesses the commercial potential. Our results indicated that synchronization protocol based on Chronogest and Folligon with the rate of success over 71%, may be useful in red deer farming. The evaluation of PAGs and basic ovarian steroids is useful tool for reproductive stage monitoring, but further diagnostics should be provided for precise determination of follicle development throughout the reproductive stage or pregnancy development in red deer females.

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
