# Peer review of "The Effectiveness of Pharmacological Synchronization of the Estrous Cycle in Hinds (Cervus elaphus L.): A Pilot Field Trial"

_animals, 2020, doi:10.3390/ani10112148_

Round 1

Reviewer 1 Report

The work covers important issues in deer farming. In general, the research was done correctly, but on a small number of animals. They should be treated as preliminary. The results are interesting and worth publishing.

Line 88 - number 12 is not after the author's name

Line 118 - it was written that the device was replaced after 7 days. It should be after 12 days

Line 147 - the treatment of semen is described quite well. However, it was not stated at what temperature it was defrosted.

In the materials and methods it was not written where exactly CIDR and sponges were placed. This was only stated in the discussion

Line 255 - number (17) should be placed after to the author

I suggest that the discussion should include the usefulness of ultrasound examinations in the control of follicular development and ovulation. The authors do not mention this method.

Did the authors control oestrus time after removing the CIDR and sponges? If they have data, you must provide them.

Author Response

The authors are thankful for an accurate and substantive review of our manuscript. All comments and suggestions are valuable for us, we agree with them.

Line 88 - number 12 is not after the author's name

AK – We corrected it.

Line 118 - it was written that the device was replaced after 7 days. It should be after 12 days

AK – We corrected it.

Line 147 - the treatment of semen is described quite well. However, it was not stated at what temperature it was defrosted.

AK – we have completed the temperature (37°C).

In the materials and methods it was not written where exactly CIDR and sponges were placed. This was only stated in the discussion

AK – we have completed it (cervix)

Line 255 - number (17) should be placed after to the author

AK – We corrected it.

I suggest that the discussion should include the usefulness of ultrasound examinations in the control of follicular development and ovulation. The authors do not mention this method.

AK – We added the sentence: “Moreover, in case of cows, the usefulness of ultrasound examinations in the control of follicular development and ovulation is often practiced and we are considering it in continuation of our studies”. (L251-253).

Did the authors control oestrus time after removing the CIDR and sponges? If they have data, you must provide them.

AK - We didn`t control oestrus after removing of CIDR or sponges without Folligon treatment.

Reviewer 2 Report

  1. Line 118: the author writes the device was replaced after 7 days. That means that totally 4 devices were used for individuals in Group I?
  2. Line 190: because not every individual pregnant, only two based on the PAG concentration and only one based on newborn in Group 1. Whether pregnant or not affects the P and E concentration in the same group. Therefore, how the authors do this statistic? The same as E2 analysis.
  3. Line 214:the author should provide the detail data for the PAG concentration or other similar data. And show it as in the methods part Line 180
  4. Line 217: “Although we observed that not all CIDR insert was deposited in the cervix before insert change procedure or it directly before its removing from cervix in all individuals from Group I.” This sentence is not clear.

Author Response

The authors are thankful for an accurate and substantive review of our manuscript. All comments and suggestions are valuable for us, we agree with them.

Line 118: the author writes the device was replaced after 7 days. That means that totally 4 devices were used for individuals in Group I?

AK – We apologize for the mistake, we corrected it – it shoul be after 12 days

Line 190: because not every individual pregnant, only two based on the PAG concentration and only one based on newborn in Group 1. Whether pregnant or not affects the P and E concentration in the same group. Therefore, how the authors do this statistic? The same as E2 analysis.

AK - Non-pregnant hinds (as showed by PAG determination): 2 individuals in Group I and 5 individuals in Group II were excluded during analyses of P4 and E2 level results from PREGNANT subgroup. The method of statistical analysis is described in Lines:178-187.

Line 214:the author should provide the detail data for the PAG concentration or other similar data. And show it as in the methods part Line 180

AK - We decided not include the detail data for PAGs because the border of absorbancy in case of this kit is 0,3. Thus these samples below 0,3 are considered as non-pregnant females, whereas those above 0,3 as pregnant hinds. The graph of results or table would be useless in our opinion.

Line 217: “Although we observed that not all CIDR insert was deposited in the cervix before insert change procedure or it directly before its removing from cervix in all individuals from Group I.” This sentence is not clear.

AK – We corrected this sentence.

Reviewer 3 Report

After reading the manuscript, one question has been raised. Does this have commercial potential?

Although I think it would be instructive if the authors put in evidence (on the conclusion) how much Chronogest and Folligon are better than CIDR and Folligon. I think it's possible to show the success rates.

Author Response

The authors are thankful for an accurate and substantive review of our manuscript. Your comments and suggestions are valuable for us, we agree with them.

After reading the manuscript, one question has been raised. Does this have commercial potential?

Although I think it would be instructive if the authors put in evidence (on the conclusion) how much Chronogest and Folligon are better than CIDR and Folligon. I think it's possible to show the success rates.

AK - Such sentences we added in Conclusion: “Artificial insemination using cryopreserved semen from valuable bulls and possibility of hind` herd synchronization without the presence of bull possess commercial potential. Our results indicated that synchronization protocol based on Chronogest and Folligon with the rate of success 71 %, may be useful in red deer farming.”